# Screening of the Maturity Status of the Tibial Tuberosity by Ultrasonography in Higher Elementary School Grade Schoolchildren

**DOI:** 10.3390/ijerph16122138

**Published:** 2019-06-17

**Authors:** Maiko Ohtaka, Izumi Hiramoto, Hiroshi Minagawa, Masashi Matsuzaki, Hideya Kodama

**Affiliations:** 1Department of Maternity Child Nursing, Akita University Graduate School of Medicine and Faculty of Medicine, Akita 010-8543, Japan; ohtaka@hs.akita-u.ac.jp (M.O.); izumih@hs.akita-u.ac.jp (I.H.); 2Joto Orthopedic Clinic, Akita 010-0003, Japan; minagawa323@gmail.com; 3Akita University Graduate School of Medicine, Akita 010-8543, Japan; matsuzaki@sonic-j.com

**Keywords:** Osgood–Schlatter disease, cartilage thickness, tibial tuberosity, schoolchildren of higher elementary school grades, musculoskeletal examination

## Abstract

This study aimed to obtain screening data on the maturity status of the tibial tuberosity in schoolchildren of higher elementary school grades for risk management of Osgood–Schlatter disease (OSD). The maturity stages and cartilage thicknesses at the tibial tuberosity were determined by ultrasonography on the occasion of a school-based musculoskeletal examination for 124 grade 5–6 elementary schoolchildren, and their associations with the students’ demographic characteristics and OSD were examined. The time-dependent changes of the maturity status of the tibial tuberosity were also examined in grade 5 students (*n* = 26) by a longitudinal survey. The cross-sectional survey showed that the epiphyseal stage was reached in 89% of girls and 35% of boys. The girls who had experienced menarche (*n* = 28) were all in the epiphyseal stage and had a decreased cartilage thickness (*p* = 0.004, after adjusting maturity stages). Students with OSD (*n* = 5) were all girls in the epiphyseal stage, and only two of them had an increased cartilage thickness. During the longitudinal survey, a marked increase in cartilage thickness from the previous measurement was observed in three boys (without clinical symptoms) and a girl who newly developed OSD. Two students with OSD without chronic pain had thin cartilage. In conclusion, for schoolchildren of higher elementary school grades, the risk of OSD is higher among girls with the epiphyseal stage. Cartilage thickness may not contribute to the diagnosis of OSD, since thick cartilage is not very common in OSD. However, cartilage thickness may reflect the status of OSD.

## 1. Introduction

The knee joint is a site where sports injuries occur frequently during the growth period [1,2]. In particular, Osgood–Schlatter disease (OSD), which is a traction apophysitis from sports overuse, occurs at the anterior aspect of the tibial tuberosity where the quadriceps attaches via the patellar ligament [3,4,5]. Approximately 10% of young children during the growth period who are active in sports are affected by OSD [6,7]. Once affected, they must restrict sports activities due to knee pain for a long period, and this may decrease their performance level beyond adolescence [8]. However, early detection and conservative treatment can enable their early return to sports activities [4].

There are some anatomical reasons why OSD typically occurs during the growth period. Before the growth period, the patellar ligament is connected to the tibia through thick cartilage. When the growth spurt starts, the ossification center appears in the cartilage and then gradually fuses with the epiphysis. Accordingly, the cartilage is replaced with bone, and, at the end of the growth period, the patellar ligament attaches directly to the tibia through epiphyseal closure [9]. OSD is thought to be prevalent when structures of the tibial tuberosity become vulnerable before epiphyseal closure [10]. Therefore, it has been assumed that assessment of the maturity status of the tibial tuberosity has some role in the risk management of OSD in young children.

Ehrenborg and Lagergren (1961) [11] classified the normal maturational process of the tibial tuberosity radiographically into four stages: The cartilaginous stage, the apophyseal stage, the epiphyseal stage, and the bony stage. The risks for OSD appeared to be associated with the maturity stages of the tibial tuberosity determined by ultrasonography [12,13,14,15,16,17]. However, from a different perspective, the main event of the maturational process of the tibial tuberosity is regression of the apophyseal cartilage at the tibial tuberosity. Therefore, the thickness of the apophyseal cartilage (cartilage thickness) may simply reflect the degree of maturity of the tibial tuberosity and may have clinical implications regarding the risk for OSD, which is independent of the maturity stage. Furthermore, cartilage swelling is known to be one of the characteristic findings of OSD [18,19,20,21,22,23]. However, there have been no clear diagnostic criteria for “cartilage swelling”, probably because there have been no large-scale screening data for cartilage thickness in young children during the growth period that could be useful for the clinical application of this measure.

In Japan, the number of schoolchildren who participate in active sports activities increases rapidly once they enter junior high school. Given this background, there is increasing interest in the musculoskeletal problems of students in higher elementary school grades, so that students in the growth period can participate healthily in sports activities for a long time [24]. Therefore, prevention and early detection/treatment of OSD are important in the school-based musculoskeletal examinations in the higher elementary school grades. For that purpose, this study aimed to obtain screening data of the maturity status of the tibial tuberosity by ultrasonography in grade 5–6 elementary schoolchildren and to explore the possibility of using the data for risk management of OSD.

## 2. Hypothesis

(1) Assessment of the maturity status of the tibial tuberosity from two aspects, the maturity stage and cartilage thickness, can identify higher elementary school grade students at high risk of OSD.

(2) Assessment of cartilage thickness can contribute to the diagnosis of OSD in higher elementary school grade students.

## 3. Materials and Methods

### 3.1. Study Type

This was an observational epidemiology study that followed the STROBE (strengthening the reporting of observational studies in epidemiology) statement.

### 3.2. Subjects

At a parent/guardian meeting of grade 5–6 students at a public elementary school in Akita prefecture in Japan, an orthopedic surgeon and a nurse (first author of this article) explained the necessity for a musculoskeletal examination for prevention of sports injuries during the growth period, as well as our purpose of evaluating the maturity of the tibial tuberosity by ultrasound. Since there were no objections, a similar explanation was provided to the students later. Subsequently, informed consent of the parents/guardians for their children to participate in this study was confirmed by a signed consent form.

This study was approved by the Ethics Committee of the Akita University Graduate School of Medicine (26 November 2013; approval number 2142).

### 3.3. Musculoskeletal Examination

A musculoskeletal examination was performed in April 2014 of 130 grade 5–6 elementary schoolchildren, excluding three students who failed to participate due to poor physical condition or family reasons. The examination was performed by a six-member team consisting of two nurses (including the first author), an orthopedic surgeon, a clinical laboratory technician, a physical therapist, and a sports trainer. A questionnaire was distributed to each student’s home in advance to obtain information on current and past musculoskeletal problems of the students. First, the students were examined to determine whether basic movements (3 m gait, stepping in place, one-foot standing, squatting, arm elevation, flexion, extension, medial rotation, and lateral rotation of the arm, forward bend, and backward bend) were possible. Subsequently, an orthopedic surgeon examined the musculoskeletal systems that appeared to be problematic in individual students. Students were diagnosed with OSD when tenderness was present at the site of the tibial tuberosity, accompanied by at least one of the following pathological findings on ultrasound: Cartilage swelling, an irregular contour of the tibial tuberosity, fragmentation of the ossification center, or fluid accumulation in the infrapatellar bursa [18,19,20,21,22,23].

The lower panel of Table 1 shows the results of the musculoskeletal examinations by sex. Regarding the basic movements, squatting (with both arms in the front and heels touching the ground) and forward bend (touching the floor with the finger or hand with arms stretched out in a forward bend) could not be performed by 10%–25% of the students. All other basic movements were possible for all students. Sixteen students had chronic musculoskeletal pain in the lower limb (10 of whom had knee pain). Twenty-two students were found to have organic musculoskeletal disorders of the lower limb, including flat feet (*n* = 8), OSD (*n* = 5), anterior talofibular ligament tear (*n* = 3), knee arthritis (*n* = 2), and spastic gait, bilateral external tibial bone, genu valgum, inflammation of the femur, foot arthritis, and shoulder arthritis (*n* = 1, each). The incidence of chronic pain in the lower limb or the knee was significantly higher in girls. The five students diagnosed with OSD were all girls who belonged to a sports club, and four of these girls complained of chronic knee pain.

### 3.4. Determination of Maturity Stages and Cartilage Thickness of the Tibial Tuberosity

An orthopedic surgeon and a clinical laboratory technician examined the site of the tibial tuberosity of the students using a diagnostic ultrasound system (Noblus, Hitachi Ltd., Tokyo, Japan). As a rule, the right tibia was examined, but the left tibia was examined in three students who complained of pain only in the left knee. During the examination, the students maintained a sitting position with knees bent at 90°, and the examiners slowly moved a high-resolution linear probe (18-MHz L64) from the medial side to the lateral side to take longitudinal images of the tibial tuberosity for about 4 s. The video images were immediately stored in the memory of the ultrasound machine and reviewed later to determine the maturity stage and cartilage thickness of the tibial tuberosity of each student.

The maturity stage of the tibial tuberosity was determined according to the four-stage classification described by Ehrenborg and Lagergren (1961) [11]. The cartilaginous stage is a state in which the patellar ligament is connected to the tibia via thick cartilage. The apophyseal stage is characterized by the appearance of the ossification center in the cartilage. The epiphyseal stage is confirmed when the ossification center is attached to the epiphysis and a thin layer of insertional cartilage is present. The bony stage is established by the presence of direct attachment of the patellar ligament to the tibia with epiphyseal closure (absence of any apophyseal cartilage). First, three examiners, the researcher, orthopedic surgeon, and clinical laboratory technician, independently determined the maturity stage for each student. When the results were different, the examiners held discussions to reach a decision. Only 4 of 124 students (3%) required such discussion, and opinions were divided in terms of the appearance of the ossification center for all four cases. Although many previous studies have examined the evolution of the maturity stages [12,13,14,15,16,17], the description of the reliability of their diagnosis was extremely poor. However, Yanagisawa et al. [17] reported that the kappa value for intra-rater reliability was very high, at 0.92, whereas the kappa coefficient of inter-rater reliability was moderate (0.70).

To measure cartilage thickness, a longitudinal image that simultaneously depicted the tibial epiphysis, tibial metaphysis, and the largest apophyseal cartilage was extracted from the video of each student. Figure 1 demonstrates the method for measuring cartilage thickness by maturity stage. When the cartilage thickness is measured using the images above, how to determine the measurement point on the tibial side poses a problem, because the surface at the tibial tuberosity is uneven. Therefore, the cartilage thickness was estimated by the largest diameter of the true circle drawn in the space of the apophyseal cartilage. As for the epiphyseal stage, the circle was drawn ignoring the ossification center in the cartilage space. Since the inter-rater error of the cartilage thickness measured using this method by three examiners did not exceed 15% of the first decimal place of the measured value (mm), valid results were thought to be obtained if the analysis were performed allowing an error range of ±15% of the measured value. 

### 3.5. Longitudinal Survey

Twenty-six students (10 boys and 16 girls) who were in grade 5 at the time of the cross-sectional survey (spring of grade 5 or “grade 5 spring”) underwent the same examinations 6 months later in October 2014 (fall of grade 5 or “grade 5 fall”; in Japan, the school year starts in April) and 18 months later in October 2015 (fall of grade 6 or “grade 6 fall”), contributing to the longitudinal data on cartilage thickness. Although none of the 26 students was diagnosed with OSD in grade 5 spring, one girl was newly diagnosed with this disease in grade 5 fall.

### 3.6. Statistics

SPSS (Ver. 24.0, IBM Japan, Ltd., Tokyo, Japan) was used for statistical analysis. All analyses were done by sex because the maturity status of the tibial tuberosity differs markedly between boys and girls. Since all numerical data, including cartilage thickness, were normally distributed (Kolmogorov-Smirnov test), parametric statistical analyses were performed. For the cross-sectional survey, the differences in students’ demographic characteristics and musculoskeletal examination results among maturity stages of the tibial tuberosity were examined by one-way analysis of variance (ANOVA) for numerical data or by the chi-squared test for percentage data. If the ANOVA was significant, the Bonferroni or Games–Howell multiple comparison procedure was used to perform multiple comparisons according to the presence or absence of equal dispersion. The impacts of demographic factors and OSD on cartilage thickness were examined by analysis of covariance adjusted for tibial tuberosity maturity stages. *p* < 0.05 was considered significant for all tests. For the longitudinal survey, one-way repeated measures ANOVA was used to determine the overall difference of means of students’ demographic characteristics and cartilage thickness. Greenhouse–Geiser correlations were performed when the assumption of sphericity was violated. If the ANOVA was significant, the Bonferroni multiple comparison procedure was used as a post hoc test.

## 4. Results

Six students who were being followed due to low height or were missing data on height and body weight due to transferring schools were excluded from the analysis. Thus, the analysis ultimately included 124 students (52 boys and 72 girls) aged 10–12 years. The upper panel of Table 1 shows a summary of the demographic characteristics of the subjects who were involved in the cross-sectional survey by sex. Secondary sex characteristics (determined by the occurrence of voice change in boys or menarche in girls) appeared in 28 girls (39%) and one boy. A total of 81 students (65%) belonged to a sports club; they belonged to the club for a mean of 3.1 years (±standard deviation, ±1.8 years; range, 1–7 years), and they played sports for a mean of 3.0 days (±1.3 days; 1–6 days), and a mean of 6.8 hours (±4.7 h; 1–20 h) per week.

Table 2 shows the proportion of subjects in each tibial tuberosity maturity stage, and the associations of the maturity stages with demographic characteristics, frequency of OSD, and cartilage thickness by sex from the cross-sectional analysis of 124 grade 5–6 elementary schoolchildren. A large sex difference was present in the proportions of maturity stages; specifically, the percentage of students in the epiphyseal stage was 89% in girls and only 35% in boys (*p* < 0.001). Height and body weight increased significantly with progression in maturity stages for both boys and girls. All students who showed secondary sex characteristics reached the epiphyseal stage. All five students diagnosed with OSD were girls in the epiphyseal stage, and four girls had experienced menarche. The overall prevalence rate of OSD in the subjects was 4%. However, the prevalence rate increased to 13% when the subjects were limited to girls in the epiphyseal stage who belonged to a sports club (*n* = 40). However, when comparing the incidence of OSD at each stage without considering sex differences, no significant difference was found (cartilaginous stage, 0/24; apophyseal stage, 0/17; epiphyseal stage, 5/82 (6%); *p* = 0.272). This insignificant result is probably due to the unbalanced number of students at each maturity stage. Assuming that the proportion of the number of students at each stage is constant and the morbidity rate of OSD is 6% in the epiphyseal stage, the sample size at which a significant difference can be detected is estimated to be 210 students or more.

Cartilage thickness decreased significantly with the progression in maturity stage in both boys and girls (Table 2, *p* < 0.001). Table 3 shows the impacts of demographic factors and OSD on the cartilage thickness after adjusting the tibial tuberosity maturity stages. There were significant impacts of age, secondary sex characteristics, and sports club affiliation on cartilage thickness. Advanced age (12 years old) and development of secondary sex characteristics were associated with decreased cartilage thickness, whereas belonging to sports clubs was associated with increased cartilage thickness. Based on these results, the cartilage thickness of girls in the epiphyseal stage without OSD was compared between those who had experienced menarche (*n* = 24) and those who had not (*n* = 36), and it was found that girls who had experienced menarche had significantly decreased cartilage thickness (4.3 ± 1.2 mm vs. 5.0 ± 0.8 mm, *p* = 0.004). The impact of age and sports club affiliation on cartilage thickness became insignificant when the comparison was performed after adjusting both the tibial tuberosity maturity stages and development of secondary sex characteristics. The incidence of students belonging to sports clubs was significantly decreased in students who developed secondary sex characteristics (14/29 vs. 67/95, *p* = 0.030).

Figure 2 shows a scatter plot of the cartilage thickness of girls with OSD (*n* = 5), together with the mean ± standard deviation of the cartilage thickness in each maturity stage for girls. Of the girls with OSD, two had a cartilage thickness that was greater than the mean and standard deviation of girls in the epiphyseal stage. The girl who had the smallest cartilage thickness did not complain of chronic knee pain.

Table 4 shows the demographic characteristics, presence of OSD, tibial tuberosity maturity stages, and cartilage thickness at each examination time point of the longitudinal study in 26 higher elementary grade schoolchildren by sex. Cartilage thickness decreased significantly along with the progression in maturity stage in boys (*p* < 0.001). For boys, the maturation stage was the cartilaginous stage in nine students at grade 5 spring, but five transitioned to the apophyseal stage, and two transitioned to the epiphyseal stage by grade 6 fall (two students remained in the cartilaginous stage). For girls, all students reached the epiphyseal stage by grade 6 fall. Figure 3 shows the percentages of each maturity stage at each time point and the changes in cartilage thickness for each student over time in the 26 students who were involved in the longitudinal survey by sex. Individual differences in the time-dependent changes in cartilage thickness were greater in boys than in girls. We focused on increases of ≥20% because the inter-rater error was <15%, and it was therefore concluded that an increase of ≥20% would indicate a definite increase. Although three boys consecutively showed ≥20% increases in cartilage thickness from the previous measurement point during observation, these students did not have clinical symptoms. In girls, cartilage thickness tended to be stable in most students by grade 6 fall. However, as for the girl who developed OSD in grade 5 fall, cartilage thickness increased from the previous measurement point (grade 5 spring) and decreased by grade 6 fall, along with the disappearance of chronic pain. The ultrasound images at each measurement point of the girl are shown in Figure 4.

## 5. Discussion

The maturity stage of the tibial tuberosity in girls during the growth period was reported to progress approximately two years earlier than in boys [11,16,25]. Consistent with these reports, the percentage of grade 5–6 elementary schoolchildren in the epiphyseal stage from the cross-sectional survey was markedly higher in girls. The cartilage thickness decreased significantly with progression in the maturity stage, and, after adjusting the maturity stage, advanced age, development of secondary sex characteristics, and sports club affiliation were associated with cartilage thickness. However, the impact of age and sports club affiliation on cartilage thickness became insignificant after adjusting both the tibial tuberosity maturity stages and the development of secondary sex characteristics. Therefore, in the same stage, the development of secondary sex characteristics is considered the single demographic factor that affects cartilage thickness. All girls who had experienced menarche were in the epiphyseal stage and had significantly decreased cartilage thickness. Considering these findings together, the regression in the apophyseal cartilage at the tibial tuberosity in girls is thought to be accelerating when girls reach the epiphyseal stage and experience menarche, probably due to changes in the endocrine environment [26,27]. This phenomenon is also predicted to occur in boys, although the present study could not confirm it due to the lack of boys in the epiphyseal stage after the appearance of secondary sex characteristics. Conventionally, the risk of developing OSD has been thought to be greatest during the apophyseal stage [10,14,15,28], although recent studies have reported the high-risk stage to be the epiphyseal stage [16,17]. The present findings support the latter observation, because the students who had OSD were all girls in the epiphyseal stage.

The 18-month longitudinal survey of cartilage thickness in grade 5 elementary school students demonstrated that almost all girls reached the epiphyseal stage, whereas more than half of boys were still in the apophyseal or cartilaginous stage in the second half of elementary school grade 6. The cartilage thickness did not necessarily decrease unidirectionally towards epiphyseal closure, and a relatively large increase in cartilage thickness was observed in three boys. The boys did not have any clinical symptoms of the knee, suggesting that this increase was a physiological phenomenon. All three boys were in the cartilaginous stage. Therefore, the thickening of chondrocytes and neovascularization around the ossification center that occur during transitioning from the cartilaginous stage to the apophyseal stage [29,30] may explain the increase in cartilage thickness during the normal maturational process.

For chronic knee pain during the growth period, a diagnosis of OSD requires differentiation from other diseases, such as osteochondrosis [31]. Although several findings including cartilage swelling, an irregular contour of the tibial tuberosity, fragmentation of the ossification center, and fluid accumulation in the infrapatellar bursa are considered characteristic of OSD [18,19,20,21,22,23], these findings were sometimes observed in asymptomatic children and are not specific particularly to OSD [20,25]. Therefore, diagnostic criteria for OSD based on ultrasound imaging have not yet been established. Cartilage swelling was reported in about 60% of patients with OSD [18,20]. Cartilage swelling is thought to occur in OSD due to edematous changes of the inflamed cartilage and/or due to neovascularization in the cartilage [14]. A notable increase in cartilage thickness was observed in a girl in the epiphyseal stage who newly developed OSD during the present longitudinal observation. If increased cartilage thickness is included in the diagnostic criteria of OSD, the finding must be determined based on the maturity stage, because the cartilage is usually thick in the cartilaginous or the apophyseal stages. However, it may not be appropriate to add the increased cartilage thickness to the diagnostic criteria because thick cartilage was observed in only two of the five students with OSD in the cross-sectional survey. However, cartilage thickness may reflect the status of OSD; Hirano et al. (2002) [32] conducted a longitudinal observation with magnetic resonance imaging (MRI) and reported that edematous changes at the tibial tuberosity were observed during the early stage of OSD. Therefore, the increased cartilage thickness of the students with OSD may indicate that the disease is in the critical stage. On the other hand, the decreased cartilage thickness of the students with OSD may indicate that the disease is near healing, as observed in the two students with OSD who did not complain of chronic pain.

OSD occurs commonly between the ages of 10 and 15 years in boys and 8 and 11 years in girls, indicating that it occurs about two years earlier in girls than in boys, similar to the progression of the maturity stage of the tibial tuberosity [4,20,23]. Therefore, risk management for OSD in grade 5–6 elementary schoolchildren should be sex-specific. Most boys are in stages prior to the time period in which the risk for developing OSD increases. Therefore, it is important to make students understand the potential risks and necessity of early treatment if they develop symptoms of OSD in the future. On the other hand, most girls are at a time when OSD occurs commonly. Based on the results of the present study, girls in the epiphyseal stage are at high risk for OSD, and careful management is necessary for students with OSD, as well as those who are suspected of having it [33,34]. However, when thin cartilage is observed in such students, this suggests that the disease is in the healing stage.

Finally, several limitations of the present study warrant discussion. First, although this study included grade 5–6 elementary school students, it lacked data pertaining to the cartilaginous stage and apophyseal stage for girls and the epiphyseal stage after the appearance of secondary sex characteristics for boys. A survey involving a wider age range may have clarified the association of the maturity status of the tibial tuberosity with the risk of OSD. Second, the number of students with the cartilaginous or apophyseal stage was much smaller than the number of students with the epiphyseal stage, and, thus, the tibial tuberosity maturity stage groups were not balanced. This weakens the effectiveness of the statistical analysis. Third, angiogenesis in the cartilage was considered a possible reason for cartilage swelling and may be indicative of the critical stage of OSD [14]. Since angiogenesis in the cartilage can be observed by Doppler ultrasound, this could potentially be verified.

## 6. Conclusions

In conclusion, the results of this study partially supported our hypothesis. Assessment of the maturity stage is useful to predict the risk of OSD. The risk is suggested to be higher for a girl who engages in active sports activity and when her maturity stage reaches the epiphyseal stage. Therefore, a girl in this situation should be careful when daily exercise results in excessive loads on her knees. Assessment of cartilage thickness may not contribute to the diagnosis of OSD, since thick cartilage is not very common in OSD. However, cartilage thickness may reflect the status of OSD. A large-scale survey to confirm these conclusions is necessary for the future.

## Figures and Tables

**Figure 1 ijerph-16-02138-f001:**
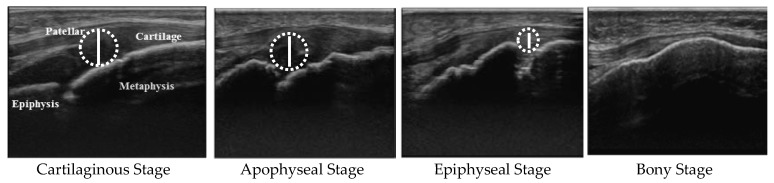
Measurement of cartilage thickness at the tibial tuberosity for each maturity stage. The cartilage thickness at the tibial tuberosity is estimated by the largest diameter of the true circle drawn in the space of the apophyseal cartilage.

**Figure 2 ijerph-16-02138-f002:**
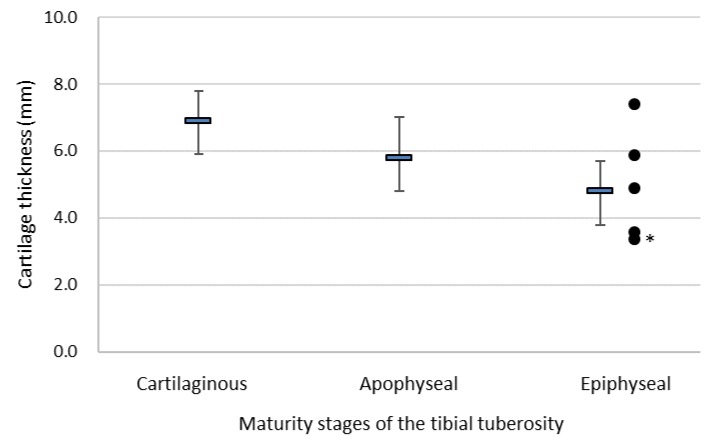
Scatter plot of the cartilage thickness of the tibial tuberosity of girls with Osgood–Schlatter disease (OSD) (●, *n* = 5) together with the mean ± standard deviation of the cartilage thickness in each maturity stage for girls. * The girl did not complain of chronic knee pain.

**Figure 3 ijerph-16-02138-f003:**
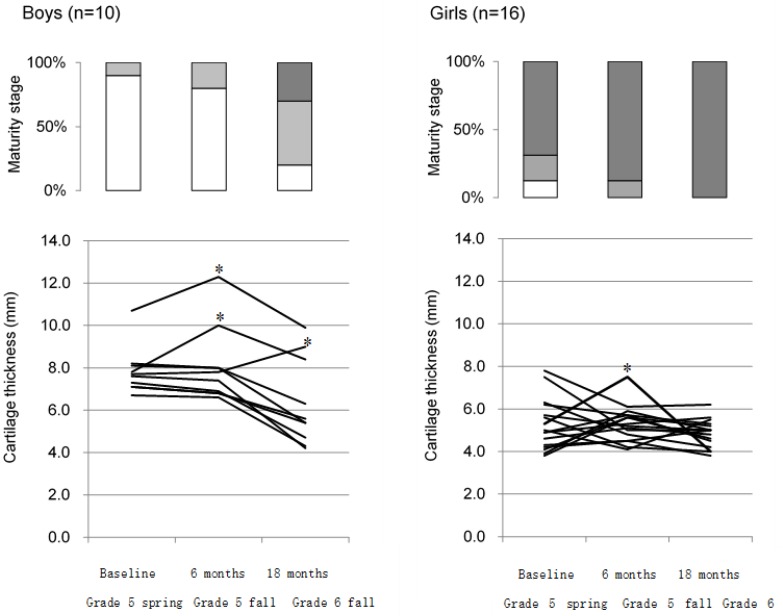
Percentage of students in each maturity stage and changes in cartilage thickness of each student at each examination time point from the longitudinal survey of grade 5 students (*n* = 26) by sex. Maturity stage (upper panel); Cartilaginous stage 
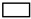
, Apophyseal stage 
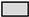
, Epiphyseal stage 
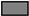
. * Measurement points showing ≥20% consecutive increases in cartilage thickness from the previous measurement during observation. A girl who showed ≥20% consecutive increases in cartilage thickness in grade 5 fall developed OSD.

**Figure 4 ijerph-16-02138-f004:**
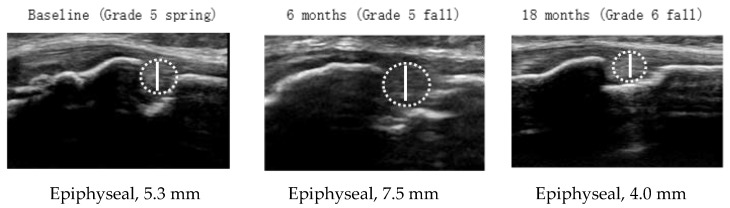
Ultrasound image, maturity stage, and cartilage thickness at each examination time point of the student who developed Osgood–Schlatter disease in grade 5 fall. Cartilage swelling and an irregular contour of the tibial tuberosity are observed in grade 5 fall. However, these findings are not observed in grade 6 fall. Chronic pain had disappeared by grade 6 fall.

**Table 1 ijerph-16-02138-t001:** Demographic characteristics and results of a musculoskeletal examination of 124 grade 5 and 6 elementary school children involved in a cross-sectional survey.

Profiles of Subjects		Boys (*n* = 52)	Girls (*n* = 72)
Demographic characteristics			
	Age (year)	10.9 ± 0.7	10.9 ± 0.8
	Height (cm)	144 ± 6	147 ± 8
	Weight (kg)	36.5 ± 5.4	39.3 ± 7.9
	Appearance of secondary sex characteristics *	1 (2)	28 (39)
	Sports club affiliation	35 (69)	46 (64)
Results of a musculoskeletal examination			
	Basic movements		
	Inability to squat	5 (10)	14 (19)
	Inability to bend forward	13 (25)	7 (10)
	Musculoskeletal problems		
	Chronic pain (systemic)	2 (4)	16 (22)
	Chronic pain (knee)	1 (2)	9 (13)
	Musculoskeletal disorder (systemic)	10 (19)	14 (19)
	Osgood–Schlatter disease	0	5 (7)

Values are means ± standard deviation or numbers of students (%). * Determined by the occurrence of voice change in boys or menarche in girls.

**Table 2 ijerph-16-02138-t002:** Proportion of subjects (124 grade 5–6 elementary schoolchildren) in tibial tuberosity maturity stages, as well as demographic characteristics, frequency of Osgood–Schlatter disease, and cartilage thickness at the tibial tuberosity of subjects in each maturity stage by sex.

**Boys**	**Cartilaginous Stage ^a^**	**Apophyseal Stage ^b^**	**Epiphyseal Stage ^c^**	**Bony Stage**	***p*-Value ^#^**	**Multiple Comparisons ^##^**
***n* = 52 (100)**	**22 (42)**	**12 (23)**	**18 (35)**	**0**		
Demographic characteristics						
Age (years)	10.5 ± 0.6	11.3 ± 0.6	11.3 ± 0.5	-	<0.001	a < b **, a < c **
Height (cm)	139 ± 4	145 ± 3	150 ± 5	-	<0.001	a < b **, a < c **, b < c **
Weight (kg)	33 ± 4	34 ± 5	40 ± 7	-	<0.001	a < b *, a < c **, b < c *
Appearance of voice change	0	0	1/18 (6)	-	0.382	
Sports club affiliation	11/22 (50)	9/12 (75)	15/18 (83)	-	0.067	
Osgood–Schlatter disease	0	0	0	-	-	
Cartilage thickness (mm)	8.1 ± 1.4	6.2 ± 1.2	5.3 ± 0.7	-	<0.001	a > b **, a > c **
**Girls**	**Cartilaginous Stage ^a^**	**Apophyseal Stage ^b^**	**Epiphyseal Stage ^c^**	**Bony Stage**	***p*-Value ^#^**	**Multiple Comparisons ^##^**
***n* = 72 (100)**	**2 (3)**	**5 (7)**	**64 (89)**	**1 (1)**		
Demographic characteristics						
Age (years)	10.0 ± 0.0	10.2 ± 0.4	11.0 ± 0.8	11	0.013	a’< c’ **, b’ < c’ *
Height (cm)	132 ± 4	136 ± 2	148 ± 7	154	0.008	b’ < c’ **
Weight (kg)	29 ± 0	29 ± 2	40 ± 7	53	<0.001	a’ < c’ **, b’ < c’ **
Appearance of menarche	0	0	27/64 (42)	1/1(100)	0.097	
Sports club affiliation	2/2 (100)	4/5 (80)	39/64 (61)	1/1(100)	0.475	
Osgood–Schlatter disease	0	0	5/64 (8)	0	0.880	
Cartilage thickness (mm)	6.9 ± 0.9	5.8 ± 1.2	4.8 ± 0.9	0	0.002	a > c *

Values are mean ± standard deviation or number of students (%). ^#^ by one-way analysis of variance for numerical data or by the chi-squared test for percentage data. ^##^ by Bonferroni: _a, b, c_ or by Games-Howell: _a’, b’, c’_; ** *p* < 0.01, * *p* < 0.05.

**Table 3 ijerph-16-02138-t003:** Impact of demographic factors and Osgood–Schlatter disease on cartilage thickness in 124 grade 5–6 elementary schoolchildren.

Variables		*n*	Cartilage Thickness (mm)	*p*-Value **
Sex				
	Males	52	5.9 ± 1.2	
	Female	72	5.5 ± 1.0	0.060
Age				
	10 or 11 y	94	5.7 ± 1.1	
	12 y	30	5.2 ± 1.0	0.013
Height				
	≥145cm	57	5.6 ± 1.2	
	<145cm	67	5.7 ± 1.0	0.846
Weight				
	≥37kg	59	5.9 ± 1.1	
	<37kg	65	5.4 ± 1.0	0.057
Secondary sex characteristics *				
	None	95	5.7 ± 1.0	
	Present	29	5.1 ± 1.1	0.009
Sports club affiliation				
	Belonged	81	5.8 ± 1.1	
	Not belonged	43	5.3 ± 0.9	0.021
Osgood–Schlatter disease				
	No	119	5.6 ± 1.1	
	Yes	5	6.1 ± 1.5	0.321

Values are means ± standard deviation. * Determined by the occurrence of voice change in boys or menarche in girls. ** Analysis of covariance, adjusted for tibial tuberosity maturity stages.

**Table 4 ijerph-16-02138-t004:** Demographic characteristics, Osgood–Schlatter disease, tibial tuberosity maturity stages, and cartilage thickness at the tibial tuberosity at each examination time point of the longitudinal study in 26 higher elementary grade schoolchildren by sex.

**Boys (*n* = 10)**	**Base Line ^a^** **Grade 5 Spring**	**6 Months ^b^** **Grade 5 Fall**	**18 Months ^c^** **Grade 6 Fall**	***p*-Value ^#^**	**Multiple Comparisons ^##^**
Demographic characteristics					
Age (y)	10.1 ± 0.3	10.5 ± 0.5	11.6 ± 0.5	<0.001	a < b *, a < c **, b < c **
Height (cm)	138 ± 4	140 ± 4	147 ± 5	<0.001	a < b **, a < c **, b < c **
Weight (kg)	32.8 ± 4.9	34.3 ± 5.1	37.8 ± 5.1	<0.001	a < b **, a < c **, b < c **
Appearance of voice change	0	0	0		
Sports club affiliation	6 (60)	6 (60)	7 (70)		
Osgood–Schlatter disease	0	0	0		
Tibial tuberosity maturity stages					
Cartilaginous stage	9 (90)	8 (80)	2 (20)		
Apophyseal stage	1 (10)	2 (20)	5 (50)		
Epiphyseal stage	0	0	3 (30)		
Bony stage	0	0	0		
Cartilage thickness (mm)	7.8 ± 1.1	8.1 ± 1.8	6.3 ± 2.0	<0.001	a > c **, b > c **
**Girls (*n* = 16)**	**Base Line ^a^** **Grade 5 Spring**	**6 Months ^b^** **Grade 5 Fall**	**18 Months ^c^** **Grade 6 Fall**	***p*-Value ^#^**	**Multiple Comparisons ^##^**
Demographic characteristics					
Age (y)	10.1 ± 0.3	10.5 ± 0.5	11.6 ± 0.5	<0.001	a < b **, a < c **, b < c **
Height (cm)	139 ± 5	142 ± 5	149 ± 4	<0.001	a < b **, a < c **, b < c **
Weight (kg)	32.5 ± 5.5	34.8 ± 5.9	40.7 ± 6.0	<0.001	a < b **, a < c **, b < c **
Appearance of menarche	1 (6)	2 (13)	9 (56)		
Sports club affiliation	11 (69)	12 (75)	10 (63)		
Osgood–Schlatter disease	0	1 (6)	1 (6)		
Tibial tuberosity maturity stages					
Cartilaginous stage	2 (13)	0	0		
Apophyseal stage	3 (19)	2 (13)	0		
Epiphyseal stage	11 (69)	14 (88)	16 (100)		
Bony stage	0	0	0		
Cartilage thickness (mm)	5.3 ± 1.2	5.4 ± 0.9	4.9 ± 0.7	0.282	

Values are means ± standard deviation or numbers of students (%). ^#^ One-way repeated measures analysis of variance (ANOVA). ^##^ by Mauchly: ^a, b, c^ ; ** *p* < 0.01, * *p* < 0.05.

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
