# Peer review of "Screening of the Maturity Status of the Tibial Tuberosity by Ultrasonography in Higher Elementary School Grade Schoolchildren"

_ijerph, 2019, doi:10.3390/ijerph16122138_

Round 1

Reviewer 1 Report

The present study aimed to obtain screening data of the maturity status of the tibial tuberosity in 15 schoolchildren of higher elementary school grades for risk management of Osgood-Schlatter disease 16 (OSD). The following considerations should be taken into account:

-Introduction: A clear hypothesis is missing before your main aim,

-Methods: There is a lack of a "Study type" subsection and a guideliness for this study type should be cited and followed.

-Table 1 should be placed in the results section.

-Methods: The four-stage 132 classification described by Ehrenborg and Lagergren (1961) has been used. Please, add the reliability of this classification supported by a reference and citation.

-Methods: Longitudinal survey needs to be detailed for a better accuracy.

-Methods: Specify the used normality analyses. 

-Methods: The major flaw is the lack of a sample size calculation. This is a missing issue that should be added.

-Tibial tuberosity maturity stages groups are not balanceed. This is another major flaw of ths study.

-A multivariate analysis by multiple linear regression would be beneficial in order to predic the cartilage thickness.

-Figure 2 and are very useful.

-Discussion: There are missing limitation in your study regarding the provided comments.

Author Response

Response to Reviewer 1

Comment: Introduction: A clear hypothesis is missing before your main aim,

Response: The hypothesis was added in section 2.

Comment: Methods: There is a lack of a "Study type" subsection and a guideline for this study type should be cited and followed.

Response: A description of the study type and the guidelines followed has been provided (section 3.1).

Comment: Table 1 should be placed in the results section.

Response: Table 1 has been moved to the Results section.

Comment: Methods: The four-stage classification described by Ehrenborg and Lagergren (1961) has been used. Please, add the reliability of this classification supported by a reference and citation.

Response: The following passage regarding the reliability of ultrasonography was inserted from a previous article, and the article was cited.

From P3, l137: “Although many previous studies have performed the evolution of the maturity stages [12-17], the description of the reliability of their diagnosis was extremely poor. However, Yonagisawa et al [17]. reported that the kappa value for the intrarater reliability was very high at 0.92, whereas the kappa coefficient of the interrater reliability was moderate (0.70)."

Comment: Methods: Longitudinal survey needs to be detailed for a better accuracy.

Response: Table 4 has been added to provide details of the longitudinal survey.

Comment: Methods: Specify the used normality analyses.

Response: An explanation that normality analyses were performed using the Kolmogorov-Smirnov test has been provided.

Comment: Methods: The major flaw is the lack of a sample size calculation. This is a missing issue that should be added.

Response: The problem is obtaining a sample size that clearly shows that OSD was significantly more frequent in the epiphyseal stage. Consequently, the following passage was added.

From P5, l206: “However, when comparing the incidence of OSD at each stage without considering sex differences, no significant difference was found (cartilaginous stage, 0/24; apophyseal stage, 0/17; epiphyseal stage, 5/82 (6%); p=0.272). This insignificant result is probably due to the unbalanced number of students at each maturity stage. Assuming that the proportion of the number of students at each stage is constant and the morbidity rate of the OSD is 6% in the epiphyseal stage, the sample size at which a significant difference can be detected is estimated to be 210 students or more.”

Comment: Tibial tuberosity maturity stages groups are not balanced. This is another major flaw of this study.

Response: The following has been added to the study limitations.

From P12, l360: “Second, the number of students with cartilaginous or apophyseal stage was much smaller than the number of students with epiphyseal stage, and thus tibial tuberosity maturity stages groups are not balanced. This is considered to weaken the effectiveness of statistical analysis.”

Comment: A multivariate analysis by multiple linear regression would be beneficial in order to predict the cartilage thickness.

Response: The study data showed a very strong correlation between cartilage thickness and stage. Moreover, there were strong reciprocal correlations among important demographic characteristics (age, height, BMI, menarche). Consequently, there was a limitation with respect to the application of a multiple regression model, and valid results could not be derived.

Comment: Figure 2 is very useful.

Response: We agree that it is valuable.

Comment: Discussion: There are missing limitation in your study regarding the provided comments.

Response: The study limitation indicated above has been explained in the final paragraph of “Discussion” section.

Reviewer 2 Report

This study reports on the use of ultrasound to analyze tibial tuberosities of elementary school children and includes cross-sectional and longitudinal arms.  The study also reports on a small number of subjects with Osgood-Schlatter disease. While generally well written, with clear figures and reasonable outcome measurements, the study is hampered by trying to achieve three goals, the cross-sectional study, the longitudinal study, and the OSD study. The entire manuscript may benefit from removal of the longitudinal and OSD data and only providing a better reporting on the cross-sectional analyses.  Alternatively, the drastic limitation of the longitudinal and OSD data need to be acknowledged and the conclusions dramatically scaled back as these data are insufficient for any generalizations. 

The cross-sectional study has a decent sample size and provides good descriptive data regarding differential maturity of the tibial tuberosity in males and females. Unfortunately, the statistics for this part of the study are incomplete.  The authors stated that they performed ANOVA or Chi squared tests but did not follow up with pairwise tests to identify which stages of maturity differed for each of the characteristics. These tests are necessary for completion.  

The longitudinal study only reports on 10 boys and 16 girls, which is quite limiting to any generalizability.  Moreover, the highlighting of students the >20% increases has no apparent value and is not supported by any statistical analyses. Why was this number decided upon on what is its importance?

As for OSD, only 5 girls are reported with this disorder and their cartilage thickness is plotted and discussed. Given this low sample number and the larger difference in cartilage thickness, I fail to see the value of including these data or trying to draw any conclusions about these findings. 

Author Response

Response to Reviewer 2

Comment: While generally well written, with clear figures and reasonable outcome measurements, the study is hampered by trying to achieve three goals, the cross-sectional study, the longitudinal study, and the OSD study. The entire manuscript may benefit from removal of the longitudinal and OSD data and only providing a better reporting on the cross-sectional analyses. Alternatively, the drastic limitation of the longitudinal and OSD data need to be acknowledged and the conclusions dramatically scaled back as these data are insufficient for any generalizations.

Response: The purpose of the tibial ultrasound screening performed in this study was to determine how the results are useful in OSD risk management. Consequently, excluding the OSD data would preclude a valid examination of this question. Although the sample size in the longitudinal study was certainly small, it was observed that an increase in cartilage thickness was seen in some students even though their clinical course was normal and that the cartilage actually thickens even when the epiphyseal stage has been reached in students with OSD. Although statistical significance was not seen for these observations, they are considered valuable. Furthermore, another reviewer (Reviewer 1) saw value in the longitudinal data and requested that details be provided about the longitudinal study. Regarding the OSD data, Reviewer 1 also commented that "Figure 2 is very useful." Consequently, the longitudinal and OSD data have been left in the current revision, in accordance with the alternative proposed in the latter half of the comment. In light of the limitations of the longitudinal and OSD data, these data were used in the discussion. The fact that the conclusions are not derived statistically is mentioned unambiguously in the results and discussion.

Comment: The cross-sectional study has a decent sample size and provides good descriptive data regarding differential maturity of the tibial tuberosity in males and females. Unfortunately, the statistics for this part of the study are incomplete.  The authors stated that they performed ANOVA or Chi squared tests but did not follow up with pairwise tests to identify which stages of maturity differed for each of the characteristics. These tests are necessary for completion. 

Response: Multiple comparisons were performed for both the cross-sectional and longitudinal data, and the results have been added to the table. A description of the method used has been added to the Statistics section.

Comment: The longitudinal study only reports on 10 boys and 16 girls, which is quite limiting to any generalizability.  Moreover, the highlighting of students the >20% increases has no apparent value and is not supported by any statistical analyses. Why was this number decided upon on what is its importance?

Response: As was mentioned above, we think presenting the longitudinal data is meaningful even with the small follow-up sample size. Attention was focused on increases of ≥ 20% because the inter-rater error was < 15%, and it was therefore concluded that an increase of ≥ 20% would indicate a definite increase (P5, l255).

Comment: As for OSD, only 5 girls are reported with this disorder and their cartilage thickness is plotted and discussed. Given this low sample number and the larger difference in cartilage thickness, I fail to see the value of including these data or trying to draw any conclusions about these findings.

Response: The purpose of this study was to examine whether measuring cartilage thickness can contribute to the diagnosis of OSD. This is clearly presented as a hypothesis (section 2, page 11, line 330). Consequently, we think Figure 2 is necessary. Because Figure 2 shows large variability in the cartilage thickness of the students with OSD, we surmised that, in OSD, thickening of the cartilage occurs during a certain period of the active stage, and that the thickness begins to decrease at a relatively early stage. Although there is no statistical support for this, and verification will be needed in the future, we think that this is an important finding in examining the diagnostic significance of cartilage thickness in OSD.

Round 2

Reviewer 1 Report

Thnaks for addressing all responses to my comments. Please, hypothesis may be included at the end of the introduction (a subsection is not necessary. STROBE criteria should be cited with an adequate reference.